# Development of an In Vitro Blood Vessel Model Using Autologous Endothelial Cells Generated from Footprint-Free hiPSCs to Analyze Interactions of the Endothelium with Blood Cell Components and Vascular Implants

**DOI:** 10.3390/cells12091217

**Published:** 2023-04-22

**Authors:** Josefin Weber, Marbod Weber, Adrian Feile, Christian Schlensak, Meltem Avci-Adali

**Affiliations:** Department of Thoracic and Cardiovascular Surgery, University Hospital Tuebingen, Calwerstraße 7/1, 72076 Tuebingen, Germany

**Keywords:** human induced pluripotent stem cells (hiPSCs), endothelial cells (ECs), tissue engineering, in vitro model, vascular implants, regenerative medicine

## Abstract

Cardiovascular diseases are the leading cause of death globally. Vascular implants, such as stents, are required to treat arterial stenosis or dilatation. The development of innovative stent materials and coatings, as well as novel preclinical testing strategies, is needed to improve the bio- and hemocompatibility of current stents. In this study, a blood vessel-like polydimethylsiloxane (PDMS) model was established to analyze the interaction of an endothelium with vascular implants, as well as blood-derived cells, in vitro. Using footprint-free human induced pluripotent stem cells (hiPSCs) and subsequent differentiation, functional endothelial cells (ECs) expressing specific markers were generated and used to endothelialize an artificial PDMS lumen. The established model was used to demonstrate the interaction of the created endothelium with blood-derived immune cells, which also allowed for real-time imaging. In addition, a stent was inserted into the endothelialized lumen to analyze the surface endothelialization of stents. In the future, this blood vessel-like model could serve as an in vitro platform to test the influence of vascular implants and coatings on endothelialization and to analyze the interaction of the endothelium with blood cell components.

## 1. Introduction

Cardiovascular diseases are a group of disorders of the heart and blood vessels and the leading cause of death worldwide [1], claiming an estimated 17.9 million lives each year. In particular, an alteration in the functionality of the vascular endothelium, a monolayer of endothelial cells (ECs) lining blood vessels, is a key mediator in the development of cardiovascular diseases, such as atherosclerosis and hypertension [2]. The endothelium is responsible for transporting oxygen, nutrients, hormones, and immune cells to the various tissues of the body and for removing waste products from tissues [3]. It is in close contact with the blood and prevents thrombosis and leukocyte adhesion under healthy conditions. It responds to cytokines or bacteria by upregulating the genes involved in inflammation [4] and regulates fluid and solute transport and vascular tone. Patients with occluded, narrowed, or aneurysmal vessels require treatment with vascular implants, such as stents [5].

However, the treatment of stenosed arteries using stents leads to vascular endothelial injury at the treatment site and initiates a complex cascade of inflammatory processes that may lead to the development of in-stent restenosis (ISR). At the site of injury, the release of pro-inflammatory factors, such as interleukin-6 (IL-6), tumor necrosis factor (TNF), and interleukin-1 (IL-1), mediate the adhesion and aggregation of inflammatory cells [6]. Inflammation can stimulate the proliferation and extracellular matrix synthesis of smooth muscle cells (SMCs) in the vessel wall, which in turn may lead to ISR of the stented vessel. In addition, both the damaged endothelium and the implanted foreign material (stent) are susceptible to the activation of coagulation and platelets, leading to the deposition of platelets and fibrin, which may result in thrombus formation.

Drug-eluting stents (DES) are commonly used to prevent restenosis by inhibiting SMC proliferation [7]. Although the beneficial effects of DES reduce inflammation and restenosis, the adverse effects delay re-endothelialization and impair endothelial function [8], preventing an essential step towards successful vessel repair [9]. Therefore, novel stent materials, designs, and coatings are needed to improve the endothelialization of blood-contacting implant surfaces [10].

The development of these novel vascular stents and coatings requires innovative in vitro models, to evaluate their bio- and hemocompatibility, as well as their effects on endothelialization, so that their biocompatibility and hemocompatibility can be continuously improved. To date, computer simulations [11] or idealized vascular models without cells have often been used to test vascular implants [12].

In this study, a blood vessel-like PDMS model was established to investigate the interaction of the endothelium with blood-derived immune cells and vascular implants. To obtain autologous ECs, footprint-free human induced pluripotent stem cells (hiPSCs) were generated from adult human urine-derived renal epithelial cells (RECs) using self-replicating mRNA (srRNA), and a protocol was established to differentiate them into hiPSC-derived ECs (hiPSC-ECs) within 6 days. These cells were then used to create an endothelium in a polydimethylsiloxane (PDMS)-based blood vessel-like model and to analyze the interaction with immune cells and implanted stents.

## 2. Materials and Methods

### 2.1. Ethics Statement

Renal epithelial cells were isolated from the urine of adult healthy donors, who gave their written informed consent to participate. The study was approved by the Ethics Committee of the Medical Faculty of the University of Tuebingen (911/2018BO2). The blood sampling procedure was also approved by the Ethics Committee of the Medical Faculty of the University of Tuebingen (287/2020BO2), and all the subjects gave their written informed consent. All the experiments were performed in accordance with the relevant guidelines and regulations. Since no living animals were used in this study, ethical approval for animal testing was not required.

### 2.2. Cultivation of hiPSCs Derived from Human Renal Epithelial Cells (RECs)

Footprint-free hiPSCs were generated by reprogramming RECs isolated from 100–200 mL of urine from healthy human donors using VEE-OKSiM-GFP srRNA encoding OCT4, KLF4, SOX2, cMYC, and GFP. Transfection and reprogramming were performed according to our recent study [13]. The obtained hiPSCs were cultivated in T25 culture flasks coated with 5 µg/mL of vitronectin (Thermo Fisher Scientific, Waltham, MA, USA) using an E8 stem cell medium (Essential 8, Thermo Fisher Scientific, Waltham, MA, USA) at 37 °C with 5% CO_2_. After reaching 70% confluence, the hiPSCs were passaged at a 1:10 split ratio or seeded in vitronectin-coated 12-well plates for endothelial differentiation. After detachment, the hiPSCs were resuspended in an E8 medium containing 10 μM of ROCK inhibitor Y-27632 (Enzo Life Sciences, Lausen, Switzerland). The medium was changed after 24 h to an E8 medium without the ROCK inhibitor and daily medium changes were performed. The hiPSCs were cultivated at 37 °C and 5% CO_2_.

### 2.3. Differentiation of hiPSCs Towards ECs

For the initial endothelial differentiation, 4 × 10^4^ hiPSCs (passage 10–25) were seeded in 12-well plates, pre-coated with 0.5 mL of vitronectin (5 µg/mL) per well (day-3) and cultivated in an E8 medium at 37 °C and 5% CO_2_. The E8 medium was changed daily (1 mL/well). To initiate the differentiation (day 0), the E8 medium was exchanged to an endothelial medium 1 containing an RPMI-1640 medium (Gibco by Thermo Fisher Scientific, Waltham, MA, USA) supplemented with 1% NEAA (Gibco by Thermo Fisher Scientific, Waltham, MA, USA), 25 ng/mL of BMP4 (PeproTech, Cranbury Township, NJ, USA), and 5 µM of CHIR99021 (PeproTech, Cranbury Township, NJ, USA). The next day, the medium was replaced with a fresh medium (1.5 mL/well). After 2 days, the medium was exchanged for an endothelial medium 2 containing a DMEM/F12 medium (Gibco by Thermo Fisher Scientific, Waltham, MA, USA) with 50 ng/mL of VEGF-A (PeproTech, Cranbury Township, NJ, USA), 1% B27 supplement, 1% NEAA (both from Gibco by Thermo Fisher Scientific, Waltham, MA, USA) and 2 µM of forskolin (Abcam, Cambridge, UK). The medium was changed every other day (2 mL/well) until the completion of the differentiation on day 6. 

### 2.4. Separation of CD31^+^ Cells after Endothelial Differentiation

At the end of the endothelial differentiation (day 6), the hiPSC-ECs were selected by magnetic cell sorting using a CD31 magnetic micro-bead kit (Miltenyi Biotec, Bergisch Gladbach, Germany) according to the manufacturer’s instructions. Briefly, the hiPSC-ECs were washed with 1 mL DPBS and detached using TrypLE (Gibco by Thermo Fisher Scientific, Waltham, MA, USA). The cells were then centrifuged at 300× *g* for 3 min and resuspended in 60 µL of the endothelial medium 2. Additionally, 20 µL of an FcR blocking reagent and 20 µL of CD31 MicroBeads were added to the cells, which were further incubated for 15 min at 4 °C. Then, 1 mL of the endothelial medium 2 was added, and the cells were centrifuged again at 300× *g* for 3 min. The cell pellet was resuspended in 1 mL of the endothelial medium 2.

At the same time, a LS column was placed in the QuadroMACS separator (Miltenyi Biotec, Bergisch Gladbach, Germany) and rinsed 3× with the endothelial medium 2. Afterwards, the cell solution was applied to the column and flushed 3× with the medium to remove unlabeled cells. Following this, the LS column was removed from the QuadroMACS separator and placed into a fresh 15 mL falcon tube. Subsequently, 5 mL of the endothelial cell growth medium-2 (EGM-2) BulletKit (Lonza Basel, Switzerland) was added to the column, and CD31^+^ cells were flushed out and subsequently transferred into a vitronectin-coated (5 µg/mL) culture flask. The cells were cultivated in the EGM-2 medium supplemented with 10% human AB serum (Merck, Darmstadt, Germany). 

### 2.5. Flow Cytometry 

The cells were washed with 1 mL of DPBS, detached using TrypLE (Gibco by Thermo Fisher Scientific, Waltham, MA, USA), and the reaction was stopped by adding TNS (PromoCell GmbH, Heidelberg, Germany). The cells were centrifuged at 600× *g* for 3 min. For intracellular staining, the cells were fixed with 4% paraformaldehyde (PFA) at room temperature (RT) for 15 min. After washing with DPBS, the cells were suspended in a permeabilization buffer (DPBS containing 2% BSA and 0.2% Triton X-100), and fluorescently labeled antibodies were added at a concentration indicated by the manufacturer and incubated at RT for 45 min. For extracellular staining, the cells were resuspended in DPBS with 2% BSA, fluorescently labeled antibodies were added at a concentration indicated by the manufacturer, and the sample was incubated for 45 min at RT. The cells were then washed with DPBS containing 2% BSA or a permeabilization buffer, suspended in 500 mL of 1× BD CellFIX solution (Becton Dickinson, Heidelberg, Germany), and measured using a BD FACScan flow cytometer (Becton Dickinson, Heidelberg, Germany). For the staining, a PE-labeled mouse anti-human CD144 (VE-cadherin) antibody (Clone: REA199, Art. No.: 130-118-495, Miltenyi Biotec, Bergisch Gladbach, Germany), PE-labeled anti-human CD31 REAfinity antibody (Clone: REA730, Art. No.: 130-110-669, Miltenyi Biotec, Bergisch Gladbach, Germany), PE-labeled anti-human CD66b REAfinity antibody (Clone: REA306, Art. 130-122-966, Miltenyi Biotec, Bergisch Gladbach, Germany), FITC-labeled mouse anti-human CD34 antibody (Clone: AC136, Art. No.: 130-113-740, Miltenyi Biotec, Bergisch Gladbach, Germany), and PE-labeled mouse anti-human CD62E (E-selectin) antibody (Clone: P2H3, Art. No.: 12-0627-42, Invitrogen by Thermo Fisher Scientific, Waltham, MA, USA) were applied.

### 2.6. Immunocytochemistry

The cells differentiated in the 12-well plates were washed 2× with 1 mL of DPBS and fixed for 15 min at RT with 4% PFA. After washing with 0.5 mL of DPBS, the cells were blocked for 1 h at RT in DPBS containing 4% BSA. For intracellular staining, the cells were incubated for 1 h at RT in a permeabilization buffer with fluorescently labeled antibodies at concentrations indicated by the manufacturer. For extracellular staining, the cells were incubated in DPBS containing 2% BSA and fluorescently labeled antibodies. After incubation, the cells were washed 3× with a permeabilization buffer or washing buffer (DPBS containing 2% BSA) and then 1× with DPBS. Subsequently, the cells were mounted using a Fluoroshield mounting medium with DAPI (Abcam, Cambridge, UK). The following antibodies were used: PE-labeled mouse anti-human CD144 (VE-cadherin, Clone: REA199, Art. No.: 130-118-495, Miltenyi Biotec, Bergisch Gladbach, Germany) antibody (Invitrogen by Thermo Fisher Scientific Waltham, MA, USA), PE-labeled anti-human CD31 REAfinity antibody (Clone: REA730, Art. No.: 130-110-669, Miltenyi Biotec, Bergisch Gladbach, Germany), PE-labeled anti-human CD66b REAfinity antibody (Clone: REA306, Art. 130-122-966, Miltenyi Biotec, Bergisch Gladbach, Germany), FITC-labeled mouse anti-human CD34 antibody (Clone: AC136, Art. No.: 130-113-740, Miltenyi Biotec, Bergisch Gladbach, Germany), and PE-labeled mouse anti-human CD62E (E-selectin) antibody (Clone: P2H3, Art. No.: 12-0627-42, Invitrogen by Thermo Fisher Scientific Waltham, MA, USA). Fluorescence images were acquired using an Axiovert 135 microscope and AxioVision 4.8.2 software (Carl Zeiss AG, Oberkochen, Germany).

### 2.7. Quantitative Real-Time Polymerase Chain Reaction (qRT-PCR)

The RNA was isolated using the Aurum™ Total RNA Mini Kit (Bio-Rad, Munich, Germany), and 300 ng of RNA was reverse transcribed into complementary DNA (cDNA) using the iScript kit (Bio-Rad, Hercules, CA, USA). The primers (Table 1) for the specific amplification of the transcripts were ordered from Eurofins Genomics (Ebersberg, Germany) and used at a final concentration of 300 nM. Real-time qRT-PCR reactions were performed using IQ SYBR Green Supermix (Bio-Rad, Hercules, CA, USA) and the CFX Connect Real-Time PCR Detection System (Bio-Rad, Hercules, CA, USA). The expression of the constitutively expressed gene GAPDH (glyceraldehyde 3-phosphate dehydrogenase) served as an internal control for the amount of RNA input. The primers were designed by using the Primer-Blast tool from NCBI (accessed on 7 January 2020) [14]. Melting temperatures and self-complementarities were checked using the Oligonucleotide Properties Calculator from Northwestern University Medical School (accessed on 7 January 2020) [15].

### 2.8. Tube Formation Assay 

The wells of a 48-well plate were coated with 150 µL of Matrigel (Corning Incorporated, Corning, NY, USA) and incubated at 37 °C for 15 min to solidify the gel. Then, 1.2 × 10^5^ hiPSC-ECs (day 12 after starting the differentiation) were seeded and after 4 h of incubation at 37 °C with 5% CO_2_, images of the cells were acquired with the Axiovert 135M phase contrast microscope (Carl-Zeiss, Jena, Germany) using the phase contrast microscope.

### 2.9. TNF-α Stimulation of hiPSC-ECs

The hiPSC-ECs were detached on days 12–18 after starting the endothelial differentiation and 3 × 10^5^ cells were seeded per well of a 12-well plate coated with 0.5 mL of vitronectin (5 µg/mL). The cells were incubated overnight in the EGM-2 medium at 37 °C with 5% CO_2_. The next day, the medium was changed to the EGM-2 medium without or with 50 ng/mL of TNF-α. After 4 h, the expression of CD62E, ICAM-1, and VCAM-1 was analyzed via flow cytometry, fluorescence microscopy, and qRT-PCR. 

### 2.10. Cultivation of Human Umbilical Vein Endothelial Cells (HUVECs)

HUVECs (passage 2–5) (Promocell, Heidelberg, Germany) were seeded in T75 cell culture flasks coated with 0.1% gelatin and cultivated at 37 °C with 5% CO_2_ in the Vasculife^®^ EnGS EC culture medium (CellSystems, Troisdorf, Germany) containing the VascuLife EnGS LifeFactors Kit (Lifeline Cell Technology, Frederick, MD, USA), 50 mg/mL of gentamicin, and 0.05 mg/mL of amphotericin B (Thermo Fisher Scientific, Waltham, MA, USA). The medium was changed every 3 days. After reaching 80% confluency, the cells were detached using trypsin/EDTA (0.04%/0.03%, PromoCell, Heidelberg, Germany).

### 2.11. Isolation of Granulocytes

Human whole blood was collected from the antecubital vein of non-medicated healthy volunteers via venipuncture in monovettes preloaded with 1 IU/mL of sodium heparin (LEO Pharma Inc., Neu-Isenburg, Germany). A total of 13.3 mL of the anticoagulant blood was layered onto 18.7 mL of the Lymphoprep™ density gradient medium (Stemcell Technologies, Vancouver, BC, Canada) and then centrifuged at 340× *g* for 45 min. Afterwards, all the separated components except the granulocytes and erythrocytes, were aspirated. The granulocyte layer was carefully extracted, washed with 15 mL DPBS and centrifuged at 340× *g* for 15 min. This step was repeated once again. The remaining erythrocytes were then lysed using an erythrocyte lysis (EL) buffer from the QIAamp RNA Blood mini kit (Qiagen, Hilden, Germany) according to the manufacturer’s instructions. Briefly, the granulocytes and erythrocytes were incubated on ice in the EL buffer for 30 min. The cells were then centrifuged at 4 °C and 400× *g* for 10 min, the supernatant was aspirated, and the residual cells were washed once again with the EL buffer. This was followed by another centrifugation at 4 °C and 400× *g* for 10 min resulting in isolated granulocytes.

### 2.12. Calcein AM Labeling of the Isolated Granulocytes

For fluorescence labeling, the granulocytes were incubated in 1 mL of RPMI-1640 medium (Gibco by Thermo Fisher Scientific, Waltham, MA, USA) containing 10% fetal bovine serum (FBS, Gibco by Thermo Fisher Scientific, Waltham, MA, USA) and 250 ng/mL of calcein AM (Invitrogen by Thermo Fisher Scientific, Waltham, MA, USA) for 15 min at 37 °C. Afterwards, the granulocytes were washed twice with DPBS.

### 2.13. Static Interaction of Granulocytes with TNF-α Stimulated hiPSC-ECs

The 3.5 × 10^5^ hiPSC-ECs were seeded per well of a 12-well plate coated with vitronectin (5 µg/mL) and incubated in the EGM-2 medium at 37 °C and 5% CO_2_ for 24 h. Then, the medium was changed to an EGM-2 medium containing 50 ng/mL TNF-α, and the cells were incubated for 4 h at 37 °C with 5% CO_2_. Afterwards, the EGM-2 medium was removed and the hiPSC-ECs were incubated with 1 × 10^5^ fluorescently labeled granulocytes for 10 min at 37 °C and 5% CO_2_. Subsequently, the non-adherent granulocytes were aspirated and the cell layer was washed with 1 mL DPBS. Fluorescence images were then acquired on five pre-defined spots in the well using an Axiovert 135 microscope and AxioVision 4.8.2 software (Carl Zeiss, Jena, Germany). The attached cells were counted using ImageJ software. 

### 2.14. Fabrication of a Blood Vessel-Like Structure Embedded in Polydimethylsiloxane (PDMS)

To fabricate a blood vessel-like structure, a histological embedding mold (25 mm × 6 mm, EXACT, Norderstedt, Germany) was pierced with a sterile 20 G needle (∅ 0.9 mm) and 3 mL of SYLGARD^®^184 (Dow, Midland, MI, USA) was filled into the embedding mold until the needle was fully covered. Therefore, the catalyst and elastomer were mixed in a ratio of 1:10 and subsequently centrifuged at 1500× *g* for 10 min to remove air bubbles. To cure the PDMS, the priced mold was incubated for 60 min at 60 °C. The needle was then removed from the embedding mold and the PDMS model containing a blood vessel-like structure was peeled out of the embedding mold (see results).

### 2.15. Population of the Blood Vessel-Like Structure with hiPSC-ECs and Perfusion

Before use, the PDMS model was incubated with 70% EtOH for 1 h at RT and then rinsed 2× with the DPBS. Then, the blood vessel-like structure was coated with 10 µg/mL of vitronectin for 45 min at RT. Following this, 1 × 10^5^ hiPSC-ECs were seeded in the EGM-2 medium and incubated for 30 min at 37 °C with 5% CO_2_. Then, the construct was rotated 180 degrees and incubated again for 30 min. This procedure was repeated two more times to completely populate the lumen of the PDMS model. In the last rotation step, the model was rotated to its initial position and incubated for 24 h at 37 °C with 5% CO_2_. Subsequently, the ability of hiPSC-ECs to stably line the lumen of the PDMS model under physiological shear forces was investigated. Therefore, a perfusion pump system was designed that contained a medium reservoir connected to a 0.25 µm filter (Sigma-Aldrich, St. Louis, MO, USA) to allow sterile oxygenation. After seeding with hiPSC-ECs, using a pump (Ismatec SA, Opfikon, Switzerland), the lumen of the PDMS model was perfused with 2.3 mL/min EGM-2 medium, corresponding to a flow rate of 5 dyne/cm^2^. The setup was run with 25 mL of the EGM-2 medium for 24 h at 37 °C and 5% CO_2_. After perfusion, the PDMS model lined with hiPSC-ECs was fixed with 4% PFA and subsequently stained with a DAPI and PE-labeled mouse anti-human CD144 (VE-cadherin) antibody. Afterwards, representative images of the PDMS model were acquired first without cutting the model to obtain an overview of the endothelialization of the lumen. Subsequently, the PDMS model was cut manually into 1 mm-thick sections and analyzed via fluorescence microscopy to obtain detailed images.

### 2.16. Dynamic Interaction of Granulocytes with TNF-α Stimulated Endothelium in the PDMS Blood Vessel Model

To analyze the interaction of immune cells (granulocytes) with hiPSC-ECs, 1 × 10^5^ Ecs were seeded into the vitronectin-coated (10 µg/mL) lumen of the PDMS blood vessel model as described above and incubated overnight at 37 °C and 5% CO_2_. The next day, the Ecs were incubated in the EGM-2 medium without or with 50 ng/mL TNF-α at 37 °C and 5% CO_2_ for 4 h. Then, 1 × 10^5^ calcein AM labeled granulocytes per ml were perfused with a flow rate of 150 µL/min for 10 min over the Ecs, corresponding to a shear stress of 0.4 dyne/cm^2^. Afterwards, fluorescence images were acquired on five pre-defined spots in the PDMS blood vessel model using an Axiovert 135 microscope and AxioVision 4.8.2 software (Carl Zeiss, Jena, Germany). The attached cells were then counted using ImageJ software (v.153f51 NIH, Bethesda, MD, USA).

### 2.17. Analysis of Stent Endothelialization in the PDMS Blood Vessel Model

To investigate the endothelialization of blood-contacting implants, stents were inserted into the lumen of the PDMS blood vessel model with a diameter of 3.5 mm and a length of 25 mm. The PDMS model was first coated with vitronectin (10 µg/mL), populated with 3.5 × 10^5^ ECs as described above, and incubated overnight at 37 °C and 5% CO_2_. Self-expanding nitinol stents (Derivo 3.5 × 15 mm, Acandis GmbH, Pforzheim, Germany) with or without vitronectin (10 µg/mL) coating were then inserted under the sterile bench into the lumen of the endothelialized PDMS model. Afterwards, the PDMS model containing the stent was filled up with the medium and incubated for 24 h at 37 °C and 5% CO_2_.

To analyze the endothelialization of the surface, the lumen with the stent was fixed with 4% PFA and the cells were stained with DAPI and ActinRed. The samples were analyzed via fluorescence microscopy. Furthermore, to prepare thin sections with the cutting and grinding method, the samples were dehydrated in ascending ethanol series (50%, 70%, 80%, 96%, and 100%). The samples were then infiltrated with an ethanol (Merck, Darmstadt, Germany) and Technovit 7200 (Morphisto GmbH, Offenbach, Germany) mixture in a ratio of 1:1 for 24 h at 4 °C and then with pure Technovit 7200 at 4 °C for 24 h. Then, curing by light was performed in an EXAKT 520 light chamber (EXAKT Advanced Technologies GmbH, Norderstedt, Germany) for 10 h under white light and 10 h under blue light. The polymerized blocks containing the stents were bonded to plastic slides (Resolab GmbH, Bad Oeynhausen, Germany) with Technovit 7210 by curing with blue light for 10 min using the EXAKT 402 precision adhesive press. Subsequently, the diamond band saw EXAKT 300/310 was used to obtain sections of 50–100 μm. To further reduce the thickness, the specimens were ground and polished using the EXAKT 400CS grinding system.

### 2.18. Statistical Analyses

Data are shown as mean ± standard deviation (SD) or standard error of the mean (SEM). Paired *t*-test or one-way ANOVA for repeated measurements followed by Bonferroni’s multiple comparison test were performed to compare means. Two-tailed statistical analyses were performed using GraphPad Prism 9.4.1 (GraphPad Software, La Jolla, CA, USA). Differences of *p* < 0.05 were considered significant.

## 3. Results

### 3.1. Differentiation of Footprint-Free Generated hiPSCs into ECs

After reprogramming the adult somatic RECs into hiPSCs, the obtained cells were seeded into vitronectin-coated wells of a 12-well plate to induce endothelial differentiation. A 6-day protocol was established using a cocktail of small molecules and growth factors, followed by the separation of CD31^+^ cells and the further cultivation of these cells (Figure 1A). After 2 days of differentiation, hiPSC colonies were induced towards the ECs and transformed into a dense cell layer. Over the next 4 days, the cells developed tube-like structures (Figure 1B). The fluorescence microscopy analysis on day 6 of the differentiation (before the separation) proved the presence of CD31-expressing cells, especially in dense cell clusters of the tube-like structures (Figure 1C). Moreover, the qRT-PCR analysis revealed a significant decrease in stem cell marker expression (Nanog, Oct4, Sox2) in the hiPSC-ECs compared with the initial hiPSCs (Figure 1D). To purify the differentiated ECs, a CD31^+^ cell separation was performed on day 6 of the differentiation. Before the separation, 38.5 ± 6.5% of cells were CD31-positive (Figure 1E). After the separation, a significant increase of CD31-expressing cells (96.9 ± 0.49%) compared to the control confirmed the successful separation of CD31-positive cells. The purified CD31^+^ ECs were applied for subsequent experiments. 

### 3.2. Characterization of hiPSC-ECs

After the completion of endothelial differentiation and the separation of CD31^+^ cells at day 6 of the differentiation, the purified ECs were further cultivated until day 20 and compared with HUVECs. At day 8 (2 days after the CD31^+^ cell separation) and 12 of the differentiation, the ECs derived from hiPSCs showed a comparable cell morphology to HUVECs (Figure 2A). In addition, a qRT-PCR analysis was performed after the CD31^+^ cell separation on days 6, 12, and 20 of the differentiation to detect EC-specific markers (Figure 2B). A strong expression of CD31 was detected on day 6 of the differentiation, directly after the separation of CD31^+^ cells. The expression levels of CD31 on days 12 and 20 were comparable to the expression levels in HUVECs. The expression of CD34 and VEGFR2 decreased significantly between days 6 and 12, but the expression levels on days 12 and 20 were similar to those of HUVECs. In general, the expression levels of EC-specific markers were significantly increased compared to the initial hiPSCs. The expression of CD31 and CD34 was confirmed by the immunostaining of hiPSC-ECs on days 8 and 12 of the differentiation (Figure 2C). A reduction in CD34 expression was also detected on day 12 of the differentiation compared with the expression on day 8. Furthermore, Appendix A shows the immunostaining of hiPSC-ECs at days 6 (before the separation) and 10 with CD144 and VEGFR2. Moreover, the flow cytometry analysis demonstrated the significantly upregulated expression of the EC-specific markers, CD31, CD34, and VE-cadherin (CD144), compared with the initial hiPSCs (Figure 2D). The constant expression levels of CD31 and CD144 and a significant reduction in CD34-expressing cells between day 6 and day 20 were detected. 

### 3.3. Functional Analyses of hiPSC-Derived ECs

EC-specific functions were analyzed on day 12 after the start of endothelial differentiation. One of the most commonly used in vitro assays to model the reorganization phase of angiogenesis is the tube formation assay. The assay measures the ability of endothelial cells, plated in subconfluent density with the appropriate extracellular matrix, to form capillary-like structures (also called tubes). In vitro, the formation of tube-like structures was observed 4 h after the seeding of cells on matrigel (Figure 3A). Additionally, hiPSC-ECs were stimulated with 50 ng/mL of TNF-α to investigate the response to a pro-inflammatory stimulus and the TNF-α stimulation resulted in the increased expression of ICAM-1, VCAM-1, and CD62E compared with unstimulated ECs (Figure 3B). Moreover, a flow cytometric analysis demonstrated significantly higher numbers of CD62E-expressing cells after the stimulation with TNF-α (Figure 3C) compared to the control group. The expression of CD62E on stimulated hiPSC-ECs was also detected via fluorescence microscopy (Figure 3D). In addition, the ability of TNF-α-stimulated hiPSC-ECs to interact with immune cells was analyzed. Therefore, granulocytes were isolated from human whole blood and evaluated (Appendix A), and then incubated with hiPSC-ECs under static conditions. Significantly higher numbers of granulocytes were attached to stimulated hiPSC-ECs than to unstimulated ones (Figure 3E,F). 

### 3.4. Fabrication of a Blood Vessel-Like PDMS Model and Analysis of Endothelialization with hiPSC-ECs

To create a transparent endothelialized vascular structure, a PDMS model was fabricated using a histological embedding mold that was punctured with a 20G needle and filled with PDMS (Figure 4A). This PDMS model (Figure 4B) was connected to a circular system with a rotary pump and a medium reservoir (Figure 4C). Then, 1 × 10^5^ hiPSC-ECs were seeded into the lumen of the PDMS model and analyzed 24 h after seeding. The staining of the cells with a VE-cadherin-specific antibody showed the formation of a confluent endothelial layer with a characteristic EC morphology in the lumen of the model (Figure 4D). This endothelial layer remained intact after 24 h of perfusion at a physiological flow rate of 5 dyne/cm^2^ (Figure 4E). The images of cross-sections showed that the entire lumen of the model was covered with ECs (Figure 4F).

### 3.5. Analysis of the Interaction of Immune Cells with the Generated Endothelium in the Blood Vessel-Like PDMS Model

To evaluate the applicability of the designed model for the analysis of the interaction with immune cells, 1 × 10^5^ hiPSC-ECs (from day 12 of the differentiation) were seeded into the lumen of the PDMS model. To mimic an inflammation, the endothelium was stimulated for 4 h with 50 ng/mL of TNF-α. After washing with DPBS, the lumen was perfused with calcein AM-labeled granulocytes (1 × 10^5^/mL) at a flow rate of 150 µL/min for 10 min. Directly afterwards images were acquired in the PDMS model to evaluate the attachment behavior (Figure 5). After 10 min, a significantly increased attachment of granulocytes to the inflamed endothelium was observed (Figure 5B) compared to the unstimulated endothelium (Figure 5A). After 10 min of perfusion, a significantly increased number of granulocytes was detected on the inflamed endothelium (Figure 5C). Images and recorded time points were also acquired after 2, 4, 6, and 8 min of perfusion in Appendix A.

### 3.6. Evaluation of the Stent Endothelialization

The endothelialization of blood-contacting implant surfaces is of great importance, since the coverage of the implant surface significantly influences the patency of the blood vessel, for example after the implantation of a stent. Thus, to evaluate the applicability of the designed model for the analysis of the interaction of the stent material with the ECs lining the blood vessel after the implantation, the diameter of the lumen of the PDMS model was increased to 3.5 mm and endothelialized with hiPSC-ECs. Vitronectin-coated and uncoated nitinol stents were implanted into the endothelialized lumen of the PDMS model. After 24 h, the cells were stained with ActinRed, and the fluorescence microscopy analysis revealed an increased migration and attachment of hiPSC-ECs and lining of the surface of the vitronectin-coated stent (Figure 6B). Uncoated stents (negative control) only rested on the dense EC-monolayer; however, the cells showed no tendency to overgrow the stent struts (Figure 6A). These results could be confirmed by generating thin sections of the implanted stents using the cutting-grinding technique (Figure 6C). Uncoated stents exhibited less endothelialization, whereas stents coated with vitronectin were much more densely endothelialized.

## 4. Discussion

The insertion of implants into blood vessels results in injury to the endothelium accompanied by an inflammatory reaction and the activation of coagulation and platelets. This leads to a risk of re-stenosis in the area of stent implantation due to the proliferation of SMCs and the activation of coagulation. Although DES can inhibit the proliferation of SMCs and reduce restenosis, this is associated with delayed healing. The absence of an endothelium on the implant material also poses the risk of artificial surface recognition and late stent thrombosis. Therefore, the development of new coatings and materials is needed to improve the compatibility of current stents. In addition to the development of new materials and coatings, their preclinical testing also plays an important role in finding the most suitable material and coating for treatment. In this study, a blood vessel-like model was developed using a PDMS material and ECs derived from hiPSCs to create a blood vessel-like in vitro model for the in vitro evaluation of the endothelialization of inserted stents and the interaction of ECs with blood-derived cells, such as immune cells.

Various strategies have been developed to fabricate in vitro vascular models, including electrospun constructs [16], 3D-printed polymeric [17] or decellularized scaffolds [18], and hydrogel-based approaches [19]. The elastomeric polymer PDMS has several interesting properties for biomedical applications, such as chemical stability, gas permeability, good mechanical properties, biocompatibility, excellent optical transparency, and ease of fabrication by molding [20,21]. Therefore, this material was used in this study to create blood vessel-like structures for endothelialization. In particular, its transparency allowed for microscopy without destroying the material and allowed for the real-time monitoring of cell adhesion as well as the visualization of cells under static and flow conditions. The created PDMS model also allowed for a controlled and easily adjustable environment for the endothelialization and implantation of a stent into an endothelialized lumen and a subsequent analysis by fluorescence microscopy and histology using thin sections obtained by the cutting and grinding method. To create a more in vivo blood vessel-mimicking model with the complex structure of a vessel, it is planned to additionally integrate SMCs and fibroblasts in a follow-up study.

To obtain adult autologous human ECs, a vascular biopsy is often required, which is an invasive procedure with a limited number of ECs obtained. HUVECs are the most widely used cell type for in vitro EC research [22,23]. However, despite their benefits, HUVECs are not patient-specific and are also limited in terms of their proliferation [24]. To obtain adult ECs in sufficient numbers, ECs derived from hiPSCs are particularly valuable, since these cells offer a nearly unlimited source for generating donor-specific ECs [25]. In this study, easily accessible urine-derived RECs were used to be reprogrammed by srRNA into hiPSCs [13]. The use of self-replicating mRNA allows for a less time-consuming, faster, and more efficient reprogramming of cells compared to reprogramming with synthetic mRNA [26]. Subsequently, a differentiation method was established to obtain autologous ECs. To increase the efficiency of the differentiation, forskolin was used since it can strongly increase the concentration of the second messenger cAMP in cells [27], which leads to the enhanced expression of VEGFR-2 [28].

An essential function of the vascular endothelium is to respond to injury and attract immune cells. Unfortunately, the stenting procedure induces extensive injury to the vascular endothelium, which may lead to complications, such as thrombogenicity, inflammation, and SMC hyperplasia [29,30], and eventually to implant failure [31]. Therefore, the stent design and material, as well as the interaction with the vascular endothelium are constantly being researched [32,33]. To date, computer simulations [11] or idealized vessel models without cells have often been applied to test vascular implants [12]. Moreover, stents have been exclusively seeded with ECs without a surrounding vessel-like structure [34], which does not adequately mimic in vivo conditions. Thus, the vascular implants are tested in in vivo animal models such as pigs [35] or dogs [36], or ex vivo models such as rabbits [37]. These models are very costly, contribute to animal suffering, and have limited translatability to the human body. Using the established blood vessel-like model, animal testing could be reduced or avoided, and the use of human cells allows for better comparability with the human in vivo system. This model could also be used to analyze the development of diseases affecting the blood vessels and the influence of growth factors or therapeutics on disease development [38]. In particular, the combination of PDMS and patient-specific hiPSCs represents a step towards personalized treatment, making the artificial model even more realistic.

## 5. Conclusions

In summary, an in vitro PDMS-based model was established to generate a blood-vessel-like model for the analysis of the interaction of the endothelium with vascular implants, as well as blood cell components, such as immune cells. Using patient-specific, footprint-free hiPSCs and subsequent differentiation, ECs with specific markers and functions were obtained and used for the endothelialization of an artificial lumen. The established blood vessel-like model allows for real-time imaging of the interaction of the created endothelium with immune cells. Furthermore, a vascular stent could be implanted into the endothelialized lumen to analyze the endothelialization of implant surfaces. In the future, this blood vessel-like model could be used to test the influence of different vascular implants and coatings on endothelialization, as well as to analyze the interaction of the endothelium with blood cells, such as immune cells. Furthermore, the use of human models for pre-clinical analyses will improve the clinical translation of novel treatment strategies.

## Figures and Tables

**Figure 1 cells-12-01217-f001:**
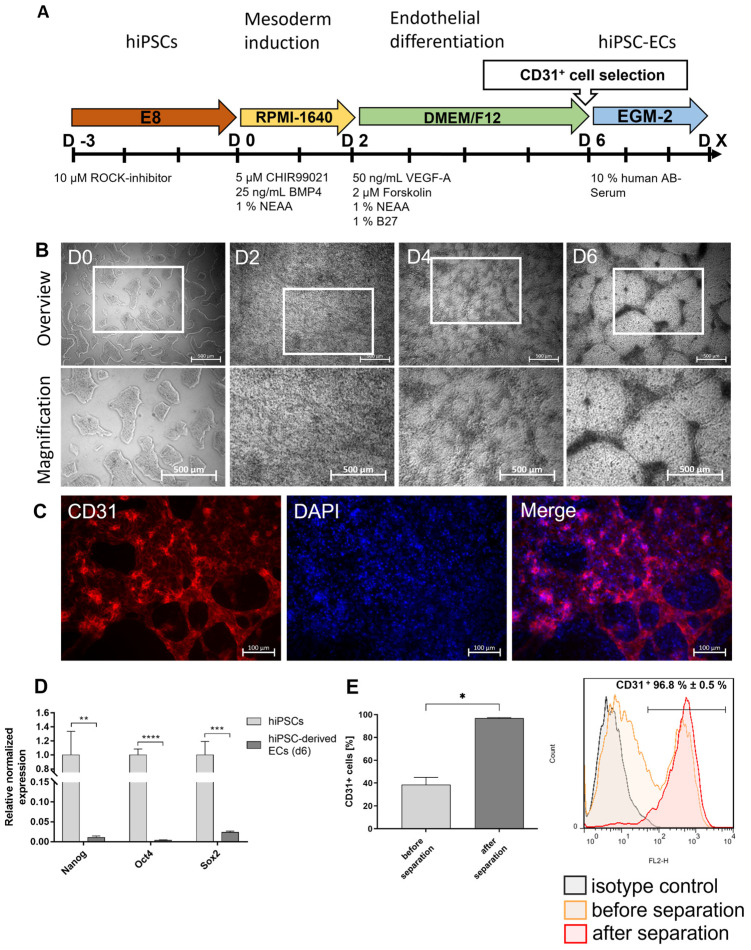
Differentiation of hiPSCs into endothelial cells (ECs). (**A**) Timeline of the protocol for the differentiation of human renal epithelial cells (RECs)-derived hiPSCs into ECs and (**B**) morphological overview and magnification of the cells at different stages cultivated in vitronectin-coated 12-well plates. Scale bars represent 500 µm. (**C**) Representative immunocytochemistry images of ECs stained with a PE-labeled CD31-specific antibody on day 6 of the differentiation (before separation). Scale bars represent 100 µm. (**D**) Expression analysis of stem cell-specific characteristics using qRT-PCR. The mRNA levels were normalized to GAPDH, and the results are presented relative to the expression levels in hiPSCs. Results are shown as mean + SEM (*n* = 3). Statistical differences were determined using one-way ANOVA. (** *p* < 0.01; *** *p* < 0.001, **** *p* < 0.0001). (**E**) Flow cytometric analysis of CD31-expressing cells before and directly after the selection of CD31^+^ cells. Results are shown as mean + SEM (*n* = 3). Statistical differences were determined using paired *t*-test (* *p* < 0.05).

**Figure 2 cells-12-01217-f002:**
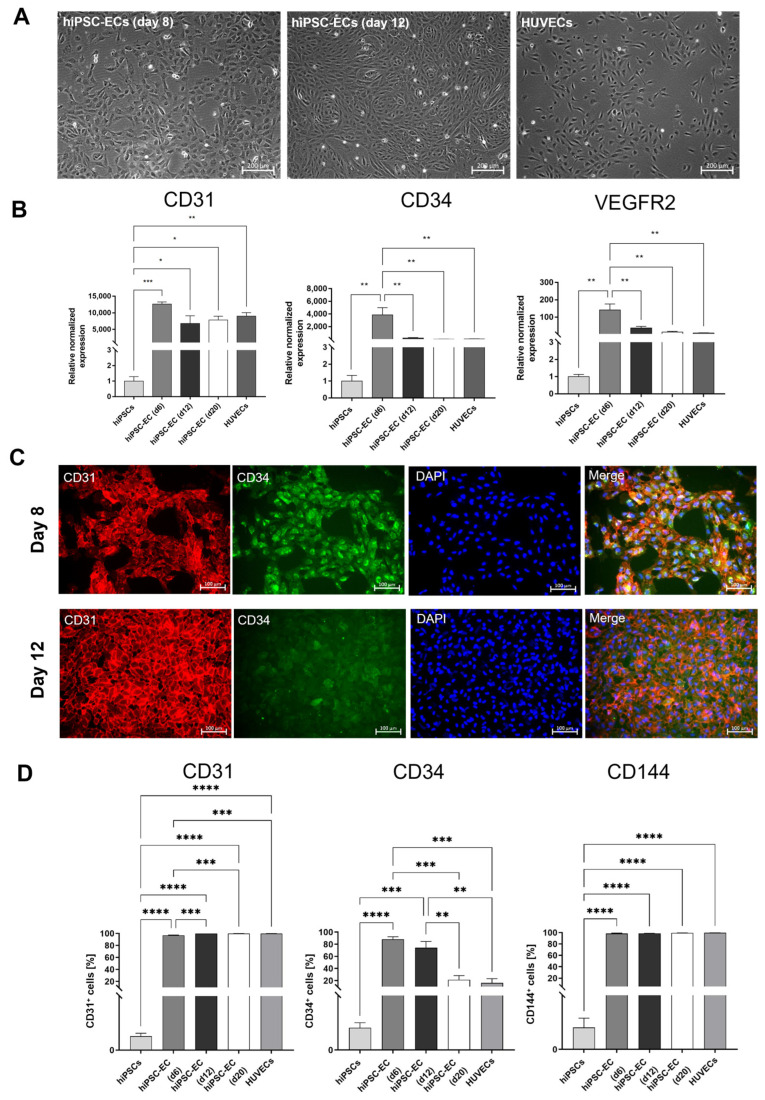
Characterization of ECs derived from hiPSCs. (**A**) Microscopic images of endothelial cells (ECs) on days 8 and 12 of the differentiation, compared with HUVECs, cultivated in vitronectin-coated 12-well plates. Scale bars of phase contrast microscopy images represent 200 µm. (**B**) Expression analysis of CD31, CD34, and VEGFR2 transcripts by qRT-PCR at different time points of the differentiation. The mRNA levels were normalized to the GAPDH mRNA levels, and the results are presented relative to the expression levels in hiPSCs. Results are shown as mean + SEM (*n* = 3). Statistical differences were determined using one-way ANOVA (* *p* < 0.05; ** *p* < 0.01; *** *p* < 0.001; **** *p* < 0.0001). (**C**) Representative immunocytochemistry images of hiPSC-derived ECs stained with PE-labeled CD31- or FITC-labeled CD34-specific antibodies on days 8 and 12 of the differentiation. Scale bars represent 100 µm. Samples were fixed, processed, and imaged identically. (**D**) Flow cytometric analysis of CD31-, CD34, and VE-cadherin (CD144)-expressing cells on days 6, 12, and 20 of the differentiation. Results are shown as mean + SEM (*n* = 3). Statistical differences were determined using one-way ANOVA (* *p* < 0.05; ** *p* < 0.01).

**Figure 3 cells-12-01217-f003:**
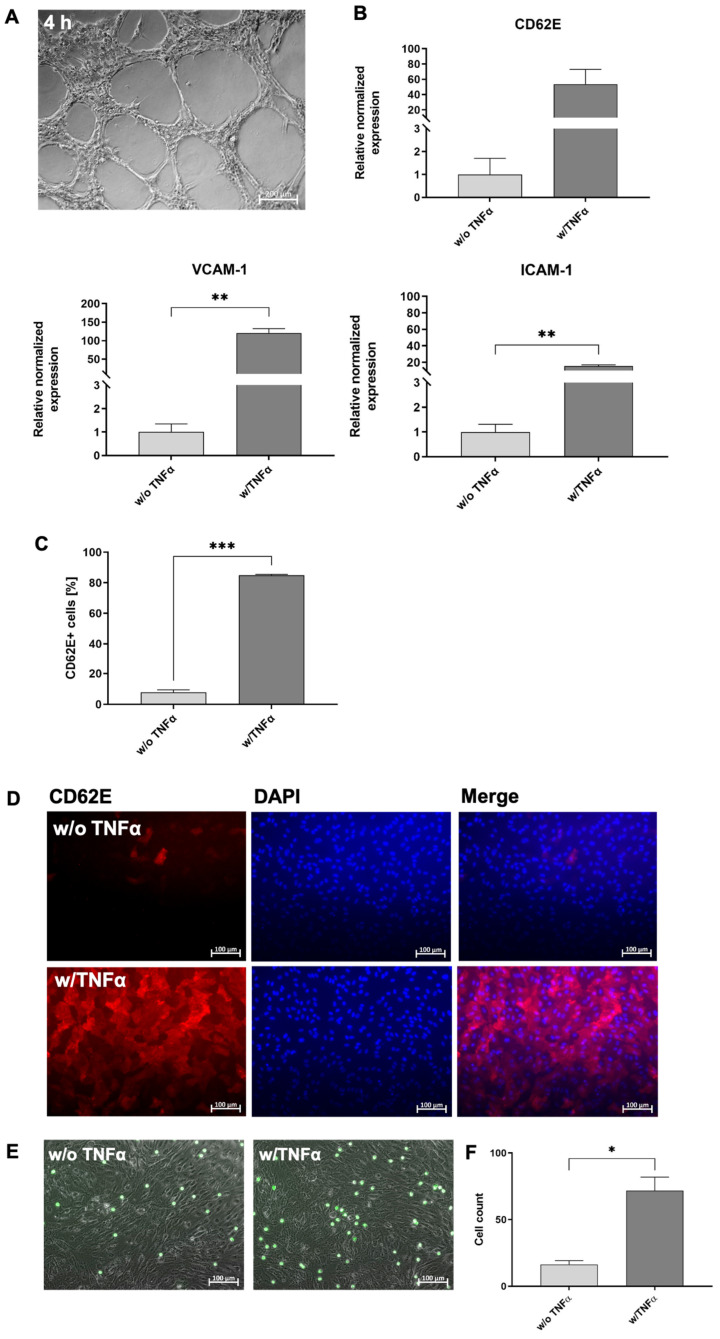
Functional analyses of hiPSCs-derived ECs (hiPSC-ECs) at day 12 of cultivation. (**A**) hiPSC-ECs formed tube-like structures when seeded on matrigel. Scale bars represent 200 µm. (**B**) Expression analysis of CD62E, ICAM-1, and VCAM-1 transcripts using qRT-PCR in unstimulated and TNF-α-stimulated hiPSC-ECs differentiated in vitronectin-coated 12-well plates. The mRNA levels were normalized to GAPDH mRNA levels, and the results are presented relative to the expression levels in hiPSCs. Results are shown as mean + SEM (*n* = 3). Statistical differences were determined using the paired *t*-test (** *p* < 0.01). (**C**) Flow cytometric analysis of CD62E-expressing hiPSC-ECs without and with 50 ng/mL of TNF-α stimulation differentiated in vitronectin-coated 12-well plates. Results are shown as mean + SEM (*n* = 3). Statistical differences were determined using paired *t*-test (** *p* < 0.01). (**D**) Representative immunocytochemistry images of hiPSC-ECs differentiated in vitronectin-coated 12-well plates without or with subsequent TNF-α stimulation and staining with a PE-labeled CD62E-specific antibody. Scale bars represent 100 µm. Samples were fixed, processed, and imaged identically. (**E**) Interaction of granulocytes (calcein AM: green) with unstimulated and TNF-α-stimulated hiPSC-ECs under static conditions. (**F**) Numbers of attached granulocytes to unstimulated and TNF-α-stimulated hiPSC-ECs under static conditions. Results are shown as mean + SEM (*n* = 3). Statistical differences were determined using the paired *t*-test (* *p* < 0.01; ** *p* < 0.01; *** *p* < 0.001).

**Figure 4 cells-12-01217-f004:**
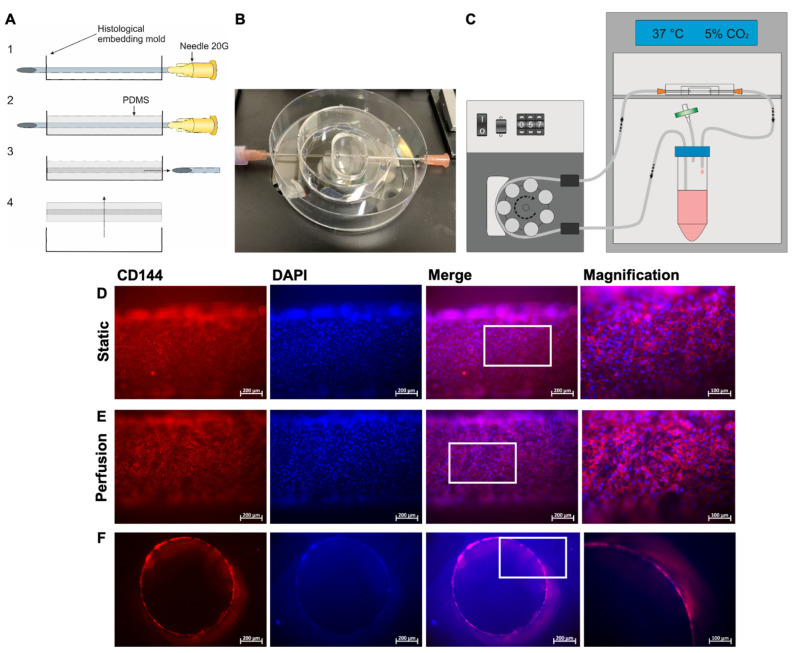
Preparation of the blood vessel-like PDMS model and colonization with hiPSC-derived ECs (hiPSC-ECs). (**A**) The PDMS model was fabricated using a histological embedding mold that was penetrated with a sterile 20G needle (1), filled up with PDMS until the needle was completely covered, and cured at 60 °C (2). Afterwards, the needle was removed (3) and the PDMS model was peeled out of the mold (4). (**B**) Representative image of a PDMS model prepared for connection to a perfusion system. (**C**) Schematic design of the experimental setup of the perfusion system. The system was placed in an incubator. (**D**) Representative immunocytochemistry images of hiPSC-ECs stained with a PE-labeled VE-cadherin-specific antibody and DAPI 24 h after seeding in the PDMS model. Representative images were acquired at different spots of the PDMS model without cutting the model. (**E**) hiPSC-ECs were seeded into the lumen of the PDMS model and cultivated for 24 h under perfusion at a flow rate of 2.3 mL/min, corresponding to a shear stress of 5 dyne/cm^2^. Representative immunocytochemistry images of hiPSC-ECs stained with a PE-labeled VE-cadherin-specific antibody and DAPI are shown. Representative images were acquired at different points of the PDMS model without cutting the model. (**F**) Representative immunocytochemistry images of ActinRed (red) and DAPI (blue) stained cross sections of the lumen of the PDMS model populated with hiPSC-ECs after 24 h of perfusion with a flow rate of 2.3 mL/min and a shear stress of 5 dyne/cm^2^. The cross-sections were generated by cutting the PDMS model manually into 1 mm thick slices. The area of the enlarged image is highlighted by the white box. Scale bars represent 100 µm or 200 µm.

**Figure 5 cells-12-01217-f005:**
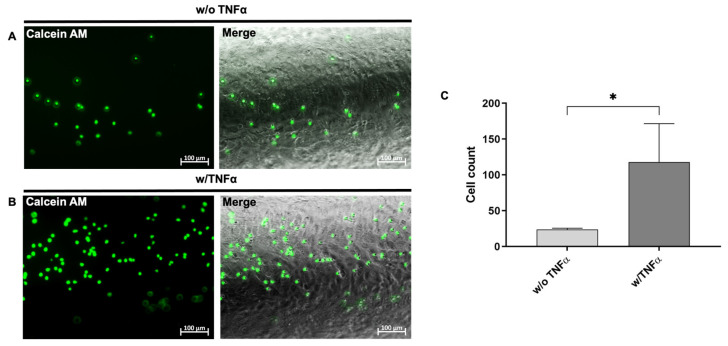
Analysis of the interaction of immune cells with the generated endothelium in the blood vessel-like PDMS model. 1 × 10^5^ calcein AM-labeled granulocytes per ml were perfused with a flow rate of 150 µL/min for 10 min over the ECs, corresponding to a shear stress of 0.4 dyne/cm². Images were acquired after 10 min of perfusion in the PDMS model. (**A**) Representative image of the interaction of calcein AM (green)-labeled granulocytes with unstimulated and (**B**) TNF-α-stimulated hiPSC-ECs in the PDMS model after 10 min of perfusion. (**C**) Numbers of attached granulocytes to unstimulated and TNF-α-stimulated hiPSC-ECs. Results are shown as mean + SEM (*n* = 3). Statistical differences were determined using the paired *t*-test (* *p* < 0.05).

**Figure 6 cells-12-01217-f006:**
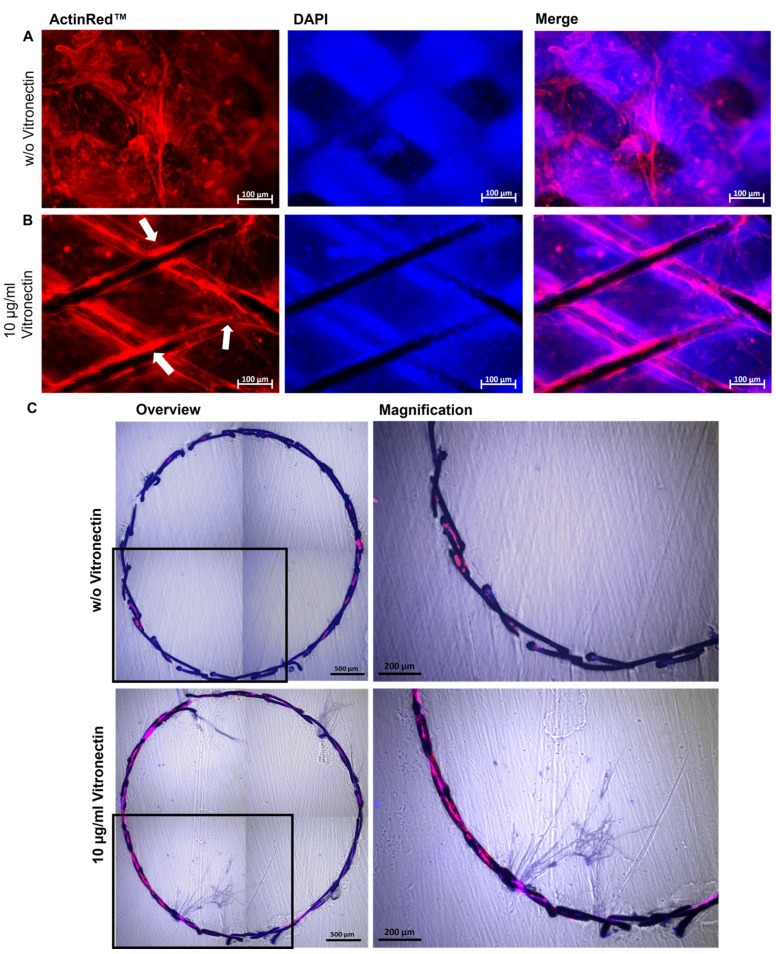
Analysis of the endothelialization of vascular stents in the blood vessel-like PDMS model. Representative immunocytochemistry images of hiPSC-ECs stained with ActinRed 24 h after the implantation of (**A**) an uncoated or (**B**) vitronectin (10 µg/mL)-coated stent into the PDMS model. Endothelialized stent struts within the PDMS model are indicated with white arrows. (**C**) Representative immunocytochemistry images of stents after applying the cutting-grinding technique: overviews (composition of four single images) and magnified sections. The hiPSC-ECs on the stent surface were stained 24 h after the implantation of an uncoated and vitronectin-coated stent into the PDMS model with ActinRed and embedded in Technovit 7200. Using a diamond band saw and grinding, 50–100 µm thick crosssections were generated. Images of the stents were composed of four single images. Scale bars represent 100 or 500 µm. (*n* = 3).

**Table 1 cells-12-01217-t001:** List of primer sequences used for the qRT-PCR analysis.

Gene	Forward Primer 5′–3′	Reverse Primer 5′–3′
**CD31**	GAACGGAAGGCTCCCTTGA	AGGGCAGGTTCATAAATAAGTGC
**CD34**	GATTGCACTGGTCACCTCGG	TCCGTGTAATAAGGGTCTTCGC
**CD62E**	GCCCAGAGCCTTCAGTGTACC	GGAATGGCTGCACCTCACAG
**GAPDH**	TCAACAGCGACACCCACTCC	TGAGGTCCACCACCCTGTTG
**ICAM-1**	CTTGAGGGCACCTACCTCTGTC	CGGCTGCTACCACAGTGATG
**Lin28**	CTTCTTCTCCGAACCAACC	CAGCCACCTGCAAACTG
**Nanog**	TGAACCTCAGCTACAAACAG	TGGTGGTAGGAGAGTAAAG
**Oct4**	AGCGAACCAGTATCGAGAAC	TTACAGAACCACACTCGGAC
**Sox2**	AGCTACAGCATGATGCAGGA	GGTCATGGAGTTGTACTGCA
**VEGFR2**	TCACAATTCCAAAAGTGATCGG	GGTCACTAACAGAAGCAATAAATGG
**VCAM-1**	ACACTTTATGTCAATGTTGCCCC	GAGGCTGTAGCTCCCCGTTAG

GAPDH: glyceraldehyde 3-phosphate dehydrogenase; ICAM-1: intercellular adhesion molecule-1; Lin28: Lin-28 homolog A; Nanog: homeobox protein NANOG; Oct4: octamer-binding transcription factor 4; Sox2: SRY-box transcription factor 2; VEGFR2: vascular endothelial growth factor 2 receptor; VCAM-1: vascular cell adhesion molecule-1.

## Data Availability

The analyzed data sets generated during the study are available from the corresponding author upon reasonable request.

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
