# Peer review of "Development of an In Vitro Blood Vessel Model Using Autologous Endothelial Cells Generated from Footprint-Free hiPSCs to Analyze Interactions of the Endothelium with Blood Cell Components and Vascular Implants"

_cells, 2023, doi:10.3390/cells12091217_

Round 1
Reviewer 1 Report
Most of the intro and discussion is well written, however the methods, results and fig legends are not up to the same standard. Recommend improving these areas, some of the particulars of which are detailed below.
Title: correct ‘Com-Ponents’ to Components
Delete lines 68-83
Methods
Please comment on how efficient the hiPSC reprogramming and differentiation processes were.
Please include clone and /or catalogue numbers for antibodies used for immunocytochemistry and flow cytometry.
Correct ‘Immunofluorescence Staining’ of cells to ‘Immunocytochemistry’, which is the correct term, throughout manuscript.
In section 2.6 please include details of slides, chamber slides or dishes used.
Looks like authors have isolated granulocytes, rather than just neutrophils (which is the largest subset of granulocytes); amend text accordingly.
Section 2.14 is not clear and requires more detail; it may be beneficial to include later diagram here to convey procedure more clearly. Also, it was not clear how stents were inserted, again diagrams or photos of process are often used to visually convey technical aspects such as this.
Justify why 0.4 dyne used for granulocyte binding assay.
Results:
Throughout results a number of images are shown, where it is not indicated whether the cells have been grown using the monolayer or matrigel culture protocol; please go through figure legends and stipulate which culture procedure was used and amend results text accordingly too.
3.1 Fig 1B does not show 3D tube like structures, as it shows a monolayer of differentiating cells.
Fig 1B please show larger images, or zooms so reader can see morphology; also is there debris at day2 & 4? If so comment accordingly.
The terms hiPSCs and hiPSC-derived ECs are used in figs, legends and text so can these also be included in culture timeline in 1A to orientate reader.
Fig 1E shows that only a subset differentiated into CD31+ cells, please comment on this, what are the remaining cells, is this representative of previous literature?
3.2 You mention VEGFR2 which is CD309 which there is no data for, I think this is meant to be CD144 which was assessed; amend text accordingly.
Wherever expression of a molecule is compared using immunocytochemistry with another treatment/time point (fig 2C & 3D); these sections need to be processed, stained, imaged and acquired in the same expt and using the same image acquisition & processing parameters and this needs to be stipulated in text or figure legend (i.e. were sections fixed at particular time points then processed together). If these images were acquired from different experiments this direct comparison cannot be made.
Fig 2 C As CD31 is not exclusive for blood endo cells, are there any images confirming CD144 protein expression? If so consider including.
Fig 2D Need to show examples of flow cytometry histogram for each marker, illustrating what has been classified as negative and positive; this can be included in supplementary data.
Fig 3A It is not apparent from this image that ECs have formed 3D tubes. This needs to be visualised using a confocal z stack to prove this. Currently it appears that cells have grown in clusters in Matrigel, with stellate cells radiating between the clumps. If the 3D tubes cannot be shown in 3D, the results and discussion text will need to be amend accordingly.
In Fig 3 legend needs to clearly indicate how cells have been cultured to generate each bit of data in this figure i.e. monolayer or Matrigel culture.
Fig 3B need flow cytometry histogram plot in sup data confirming how positive and negative cells were distinguished.
Fig 3 More info needed in fig legend and results re how expts conducted i.e. Fig 3E was this conducted under static conditions on monolayer?
3.4 Figure 4C does not clearly indicate EC morphology or the tight junctions; to visualise this need higher magnification images with better resolution. Revise images or text in section accordingly. If these images have been processed & acquired together using same settings, suggest using Image J to provide more robust quantification re VE cadherin expression level between static and perfusion.
Fig 4A is crucial to provide understanding; is the needle removed? Suggest improving this Fig, with additional step by step diagrams/photos and enlarging to convey how model is constructed.
Enlarge Fig 4B
Fig 4C &D. Detail in legend how these images are acquired; is the model sectioned and imaged, or can it be stained and visualised as a whole. This technique needs to be conveyed briefly in legend and in methods.
Fig 5. In results/methods/legend it is not detailed how these images are acquired, only later in discussion is it explained that this model can be imaged in real time, but this needs to be communicated earlier in methods and in text and legend relating to this figure.
3.5 Videos are not shown, but time points from real time imaging is, amend text accordingly.
Fig 6C the difference with and without vitronectin is not evident from these images, could this be quantified using image J or show zoom images to provide confidence in this data.
Author Response
Dear reviewer,
thank you for the helpful comments. Please find our answers to your comments in the document below.

Reviewer 2 Report
Weber et al. They present an interesting manuscript, well planned and with solid data. The experimental design is steely, although not very new. The authors make an adequate presentation of the results with quality and self-explanatory figures.
I would only advise authors to discuss formal aspects of quantification, authors should include manuscripts such as doi: 10.3390/ijms21072487. The conclusions are supported by the results.
Authors should end the manuscript with more translational statements and include a graphical summary.
Authors must correct small errors in grammar and style.
In global and formal terms, this manuscript is of interest and should be published.
Author Response

(The authors gave the same response as above.)

Round 2
Reviewer 1 Report
Most comments submitted in original review were addressed, whilst a few were not fully addressed e.g. still saying tube formation when there is no evidence of this and seeing low magCD144 expression doesn't mean that these CD144 molecules are configured to form functional tight junctions in the junctional space.
Author Response
Dear reviewer,
thank you once again for the helpful comments. Please find our answers to your comments in the document below.
